# Adulterant Detection in Peppermint Oil by Means of Holographic Photopolymers Based on Composite Materials with Liquid Crystal

**DOI:** 10.3390/polym14051061

**Published:** 2022-03-07

**Authors:** Wafaa Miloua, Manuel Ortuño, Víctor Navarro-Fuster, Augusto Beléndez, Inmaculada Pascual

**Affiliations:** 1Instituto Universitario de Física Aplicada a las Ciencias y las Tecnologías, Universidad de Alicante, P.O. Box 99, E-03080 Alicante, Spain; mos@ua.es (M.O.); victor.navarro@ua.es (V.N.-F.); a.belendez@ua.es (A.B.); pascual@ua.es (I.P.); 2Departamento de Física, Ingeniería de Sistemas y Teoría de la Señal, Universidad de Alicante, P.O. Box 99, E-03080 Alicante, Spain; 3Departamento de Óptica, Farmacología y Anatomía, Universidad de Alicante, P.O. Box 99, E-03080 Alicante, Spain

**Keywords:** holography, liquid crystal, diffraction grating

## Abstract

Diffraction gratings are recorded in a holographic photopolymer containing nematic liquid crystal and peppermint oil. The presence of the oil modifies the polymerization and the holographic response. The composite containing oil adulterated with triethyl citrate obtains a diffraction efficiency related to the oil’s purity. The results obtained suggest the possibility of developing a holographic chemical analysis method for quality control of raw materials.

## 1. Introduction

Essential oils are also known as volatile oils, ethereal oils, or simply as the oil of the plant from which they were extracted, such as the peppermint oil that we have been working on in our research. The biosynthesis of these volatile components is done in dedicated cell types present in almost all parts of the plant, from the leaves or flower to the roots depending on the plant’s genus. They are currently of growing importance due to their natural and sustainable origin, their independence from petroleum-based raw materials, and their multiple applications in the manufacture of perfumes, cosmetics, flavorings, and the production of substances with pharmacological properties [1,2].

Peppermint oil is often obtained by hydrodistillation, steam distillation, or solvent extraction of the fresh overground parts of *Mentha x piperita* L., a member of the Lamiaceae family. It is generally followed by rectification and fractionation before use. It is an essential oil that contains over 40 different compounds, including menthol, which gives peppermint it’s refreshing qualities. It is a common essential oil around the world [3]. 

Often, a commercial essential oil is adulterated to increase the natural oil’s desirable properties or avoid costly manufacturing of all-natural oil. Adulteration is usually accomplished by adding a similar but cheaper oil, such as cornmint oil (*Mentha arvensis*), or diluting the oil with various solvent oils [4,5]. Triethyl citrate is a synthetic solvent that could be used to adulterate pure peppermint oil.

In the last years, the development of novel sensors and the detection of substances has become a research area of great interest [6,7,8,9,10,11]. Holographic techniques have been used to detect various chemical substances and other parameters such as humidity or temperature [12,13,14,15].

Photopolymers are a type of holographic recording material in which a diffraction grating is recorded using a spatially localized photopolymerization producing fringes in the material with different refractive indexes. The areas exposed to light are polymerized, and their refractive index is higher than that of adjacent areas not exposed to light and therefore not polymerized [16].

These materials have balanced properties compared to other holographic recording materials. Perhaps the most important is the flexibility in the variation of its formulation to obtain specific characteristics [17]. This is very important for application in sensors; for example, hydrophilic photopolymers based on poly(vinyl alcohol) and acrylamide have been used to detect humidity and different chemical substances [18,19]. On the other hand, hydrophobic materials as holographic polymer-dispersed liquid crystals (HPDLC) have been used to detect vibration or as fingerprint sensors [20,21]. 

Recently, our research team has combined HPDLC with other substances such as tetrahydrofuran and tert-Butylthiol to obtain changes in the chemistry of the free radical polymerization mechanism that produce variations in the holographic response of the material allowing the detection/analysis of these substances [22].

In HPDLC, the areas exposed to light are polymerized, but their refractive index is lower than that of adjacent areas not exposed to light and therefore not polymerized, unlike in a pure photopolymer. This is due to the diffusion of liquid crystal to unexposed areas during polymerization and the relatively high value of the average refractive index of liquid crystal.

In this work, HPDLC is used to detect peppermint oil adulterated with triethyl citrate by variations obtained in recording a holographic diffraction grating. In order to make this, peppermint oil is introduced in the HPDLC formulation to obtain a composite whose holographic response allows the detection of potential adulterants in the oil.

## 2. Peppermint Oil Composition 

Essential oil is a concentrated hydrophobic liquid containing volatile (easily evaporated at normal temperatures) chemical compounds from the secondary metabolites of plants. Research shows the composition of the essential oils and extracts may vary according to the extraction method used [23]. Variations lie in differences in the proportion of the compounds and even differences in the number of compounds [24].

Peppermint oil contains 4.5–10.0% *w*/*w* of esters, calculated as menthyl acetate, C_12_H_22_O_2_; not less than 44 percent *w*/*w* of the free alcohols, calculated as menthol, C_10_H_20_O; and 15–32% *w*/*w* of ketones, calculated as menthone, C_10_H_18_O [25].

The structures of the main components are included in Table 1. As can be seen, these terpenes and terpenoids are functionalized hydrocarbons with double bonds, esters, ethers, and ketones. Either they do not contain hydroxyl groups, or they contain only one -OH per molecule; therefore, they are not soluble in aqueous media and are soluble in solvents of higher hydrophobicity. This suggests that this essential oil can be homogeneously mixed with the components of the HPDLC.

Molecules with hydroxyl groups could influence the initiation stage of polymerization, but the radicals formed from them would have no greater stability than those produced by the photopolymer initiator system, so their influence is expected to be negligible.

Of all the functional groups present, the ones most likely to influence a free radical polymerization are the double bonds. Molecules with one double bond can act as comonomers, while molecules with 2 or more double bonds can act as cross-linkers of the chains produced by the main monomer [26].

## 3. Obtaining Pure Peppermint Oil

For this experiment, pure and freshly obtained essential oil was used to avoid the presence of substances that are the product of hydrolysis, transformation, or degradation (oxidation) of the essential oil, which could affect the results obtained.

The peppermint oil from this study was obtained from the leaves of the perennial herb *Mentha Piperita* L. by hydrodistillation. For every kilogram of fresh mint leaves, we obtained 9.05 g of peppermint oil, equal to a 0.9% yield of fresh weight mint leaves. The Chromatographic profiling of the peppermint oil was done with gas chromatography with a flame ionization detector. The individual peaks were identified from the retention time compared with those standards; the components are shown in Table 2.

The three main components: menthol, isomenthone, and menthone (75.86% area), cannot act as a comonomer or cross-linker in the polymerization reaction due to their chemical structure; specifically, these molecules do not have any double bond. All of the minority components except eucalyptol and octanol have double bonds, i.e., about 19% maximum area. Therefore, they could act as comonomers or cross-linkers in a free radical polymerization reaction. 

## 4. Peppermint Oil Adulteration

As the prices of peppermint essential oil increase, so does the intentional reduction of its quality through fraudulent means with the overall purpose of obtaining an improvement in economic profit. In case an essential oil has been adulterated, it may still smell much like an unadulterated essential oil, but the composition has been modified, and the oil may not be suitable for its intended application.

There are different ways of adulterating an essential oil. One of them is the addition of single raw materials. This form of adulteration can be easily detected by analytical methods or not, depending on the substance in question [27].

Essential oils are characterized by their physical properties; according to International Pharmacopoeia, some of them have a relative density: 0.900–0.916, a refractive index: 1.457–1.467, and optical rotation: −10° to −30°. These are characteristic features of each essential oil, and they could change when they are diluted or mixed with other substances. 

Substances with great potential as adulterants are those that do not change the main physical parameters, act as solvents of essential oils, and have a relatively low cost. We have selected triethyl citrate as a potential adulterant for peppermint oil due to it has characteristics adequate for this use. It is a synthetic substance with a low cost; it is approved for use as a food additive, it is chemically inert under the conditions of use, and it is a good solvent for essential oils. Moreover, it is colorless, and therefore, the color of the essential oil would be almost unaltered. On the other hand, other physical properties as density or refraction index are similar to those of peppermint oil [9,28].

It is necessary to consider that peppermint oil is a natural product, and its physical parameters are not constant but vary between the limits shown in Table 3. Triethyl citrate has constant parameters different from those from the oil, but mixing with the oil in relatively low proportion produces an adulterated oil whose properties are in the intervals shown in the Table. Therefore, the adulteration would be difficult to detect with basic measurements of physical parameters. It would be necessary to perform an analysis using gas chromatography or high-performance liquid chromatography to detect the adulteration [29,30].

## 5. Photopolymer Formulation and Sample Preparation

The HPDLC is composed of dipentaerythritol penta/hexa-acrylate (DPHPA) as a monomer (refractive index *n* = 1.490), ethyl eosin (YEt) as a dye, *N*-phenyl glycine (NPG) as initiator, octanoic acid (OA) as cosolvent and surfactant, all of them obtained from Merck. The photopolymer also contains the nematic liquid crystal QYPDLC-036 from Qingdao QY Liquid Crystal Co., Ltd. (equipement source: Qingdao, China) equipement sourced from China, which is a mixture of 4-cyanobiphenyls with alkyl chains of different lengths. It has an ordinary refractive index n_0_ = 1.520 and a difference between extraordinary and ordinary index Δ*n* = 0.250. Table 4 shows the composition of the photopolymer.

The solution is prepared by mixing the components in a red light environment to which the photopolymer is not sensitive. 

The HPDLC/adulterated peppermint oil composite is prepared by mixing 200 μL of HPDLC with 50 μL of adulterated peppermint oil. The peppermint oil is mixed beforehand with triethyl citrate to obtain solutions of adulterated peppermint oil with different purity.

Five samples of HPDLC/adulterated peppermint oil composite with different purity of the peppermint oil are prepared to modify the volume of triethyl citrate added to peppermint oil (Table 5). These values have been selected in order to simulate a real adulteration. Nevertheless, excessive addition of triethyl citrate could change the organoleptic properties of the mix, and the adulteration could be easily detected.

The solution is introduced in an ultrasonic bath at 35 °C for 15 min. A volume of 16 μL of the solution is deposited between two glass plates 1 mm thick and separated with 13 µm glass hollow microspheres from Merck. The samples were exposed to a laser exposure (λ = 532 nm) in a holographic setup (Section 3) in order to record a diffraction grating in the composite.

## 6. Holographic Setup

Diffraction gratings are recorded in the HPDLC/adulterated peppermint oil composite samples using the holographic setup shown in Figure 1. A Nd:YAG continuous laser with a wavelength of 532 nm was utilized to record volume diffraction gratings. The laser beam was split into a reference beam and object beam with the same irradiance of 1 mW/cm^2^. The diameter of these beams was expanded to 10 mm using lenses, and spatial filters with pinholes were used to improve the quality and homogeneity of the beams. The object and reference beams were recombined at the sample at an angle of 16.0 degrees to the normal using mirrors. The spatial frequency obtained in the interference of the two beams was 1036 lines/mm. The diffracted and transmitted intensity were monitored in real-time with a He-Ne laser (632.8 nm) positioned at Bragg’s angle (19.1°). The ambient temperature in which recordings were made was 23 °C.

After recording, the samples are exposed to incoherent white light to fix the hologram.

With this setup, a photopolymerization reaction takes place in the exposed areas of the diffraction grating generated by the 532 nm laser beams. The He-Ne laser is used to reconstruct the hologram.

The initiator system of the HPDLC produces free radicals generated by light from the NPG that start the polymerization of the monomer DPHPA. This produces a highly reticulated polymer network due to DPHPA being a multifunctional monomer. Moreover, a photopolymerization-induced phase separation process (PIPS) takes place due to the high degree of cross-linking of the polymer network. In PIPS, the liquid crystal (LC) molecules diffuse to unexposed areas in the diffraction grating where they remain as droplets [31,32,33].

Figure 2 (left), shows a representation of the diffraction grating obtained into the sample after recording. On the right, the arrows show holes in the network with similar or higher liquid crystal molecules. 

## 7. Results

### 7.1. Compatibility of Peppermint Oil with the HPDLC

Samples exposed to the holographic recording were analyzed by scanning electron microscopy (SEM) to check that the introduction of peppermint oil in the photopolymer formulation does not affect the diffraction grating formation.

Figure 3 shows SEM images for a sample containing photopolymer with peppermint oil (purity 80%) after holographic exposure and white light fixing. To obtain these images, the samples were cut crosswise and immersed in toluene for 30 s. This process removed the liquid crystal droplets from the unexposed areas of the diffraction grating, leaving voids whose surface is coated with a thin layer of gold.

The images show that the holographic recording produces a diffraction grating in the form of stripes generated using the photopolymerization process. This is the expected result for an HPDLC. Therefore the peppermint oil does not change the overall process of obtaining the diffraction grating as an internal structure into the material [34,35,36].

### 7.2. Recording Time Selection

The HPDLC was used to select an adequate exposure time for the hologram recording. We made three recordings with exposure times: 40 s, 60 s, and 100 s. Figure 4 shows the maximum diffraction efficiency (DEmax) plus maximum transmission efficiency (TEmax) versus DEmax for the three exposure times.

An exposure time of 60 s obtains the higher value for DEmax + TEmax with a DEmax similar to that of 100 s. A high DEmax + TEmax value implies a low noise level, i.e., low light diffusion losses. Therefore, the 60 s exposure time is selected for the experiments.

### 7.3. Influence of the Triethyl Citrate Concentration in the Recording of the Diffraction Grating

We studied the influence of the triethyl citrate concentration related to the purity of the adulterated essential oil in recording the diffraction grating. Figure 5 shows the angular response curves obtained by the samples after the holographic recording and white light exposure.

Samples obtain different values for maximum diffraction efficiency (DEmax). These values are included in Table 6.

DEmax increases when the purity of the oil decreases, i.e., when the content of peppermint oil in the HPDLC is high, the DEmax obtained is low. Theoretical DEmax can be calculated with Equation (1) because the diffraction gratings stored in the composites behave as phase and volume gratings according to Kogelnik coupled wave theory [37].
(1)$$\mathrm{DEmax}=\Gamma \sin^{2}\frac{{\pi n}_{1}{\left(t\right)}_{\max}d}{\lambda \prime \cos\theta_{i}\prime }$$

In this equation *Γ* is the absorption, diffusion, and reflection losses factor. *θ*_*i*_’ is the reconstruction beam angle (Figure 1), measured into the material, which could be obtained with refraction Snell law. λ’ is the reconstruction beam wavelength. n_1_(*t*)_max_ is the maximum refraction index modulation, and *d* is the diffraction grating thickness.

DEmax of each curve in Figure 3 is related to the product n_1_(*t*)_max_ *d* with the other factors approximately constant. On the other hand, the width of the angular response curves is related to *d* that is similar for all samples (sample 3 has a somewhat larger curve width, which affects the value of *d*). Therefore, DEmax is directly related to the index modulation obtained for the photopolymer [38,39].

A low DEmax is related to a low index modulation that is related to poor diffusion of liquid crystal molecules to the unexposed areas of the diffraction grating. Therefore, the results showed an unexpected problem with the PIPS process when peppermint oil was introduced in the photopolymer formulation. This effect was higher when the oil content in the photopolymer increased, i.e., when the oil’s purity was high.

Figure 6 shows the DEmax values obtained in the holographic recording as a function of the purity of the peppermint oil.

As can be seen in the Figure, there is a linear trend between the DEmax and the purity of the oil. With the experimental equation obtained, it was possible to determine the purity of an oil sample with an unknown purity. This method could serve as a quick or preliminary analysis in quality control of peppermint oil. 

A possible explanation for this result is that the unsaturated components of the peppermint oil influence the PIPS process during the recording of the diffraction grating. The number of double bonds in these molecules is lower than in the DPHA main monomer. Therefore, they act as comonomers or cross-linkers during the polymer network formation, joining DPHPA molecules and changing the size of the holes produced in the polymer network. In Figure 2 (right), the size of the holes would be increased by the components with double bonds of the peppermint oil. This affected the diffusion of the LC during the PIPS, and thus, the quantity and size of the LC droplets placed at the unexposed zones would be modified. The quantity of LC diffused to unexposed zones affected the index modulation, and therefore, a low DEmax was obtained if relatively big holes were produced.

Figure 7 shows high-resolution SEM images for a sample with peppermint oil (purity 50%) and with a cross-sectional viewpoint for the center and right images. The SEM technique allows visualizing the structure of the diffraction network generated.

Figure 8 is an enlargement of Figure 7 right. Separating lines between exposed and unexposed areas can be seen (interface)—also, areas with a carryover of material due to the sample cutting process. The fringes are made up of pseudo-spherical clusters that, before sample processing, would be LC droplets or holes in the polymer network. 

Consecutive fringes have a different refractive index, but their structure is similar due to the sample’s processes: laser exposure, incoherent white light exposure, cutting, and gold coating for SEM. The areas exposed during the holographic recording are polymerized, and the unexposed areas are subsequently polymerized on exposure to incoherent white light. Therefore, it is not possible to visually differentiate exposed and unexposed areas. 

## 8. Conclusions

Incorporating peppermint essential oil into an HPDLC photopolymer yields a composite material that can be used as a holographic recording material. The components of peppermint essential oil influence photopolymerization by affecting the diffraction efficiency obtained. High purity of the essential oil implies lower diffraction efficiency. A possible explanation for this result has been advanced, which is related to the fact that the presence of substances with double bonds in the essential oil can make them act as comonomers or cross-linkers, modifying the size of the polymer network holes, which would have a direct influence on the diffraction efficiency obtained. The larger the size of the holes in the polymer network, the less diffusion of LC to unexposed areas (PIPS) and therefore less modulation of the refractive index and diffraction efficiency.

Furthermore, it has been obtained that the decrease in the maximum diffraction efficiency is proportional to the purity of the essential oil adulterated with triethyl citrate. This result can be used to develop a holographic method that allows identifying the purity of a peppermint oil adulterated with triethyl citrate. The experiments imply the incorporation of peppermint oil into an HPDLC to reach a concentration of about 17%. For each holographic recording, it is necessary to have 16 μL of the solution, i.e., about 3 μL of oil. As the photopolymer must be mixed with the oil by mechanical means, the quantity of solution prepared must be higher, but the method has the potential for work with small volumes of the sample. The method is not selective, and therefore adulterants other than triethyl citrate could produce the same result since the adulterant acts as a diluent of the oil, changing the concentration of the minority components with double bonds. This could be an advantage to develop a quick analysis method because the adulteration could even be detected if substances different from triethyl citrate were used. Additional work is necessary to evaluate other substances that could be used as adulterants and their influence on the holographic response.

On the other hand, other essential oils or even other kinds of natural products with molecules with double bonds could be analyzed with this method. The research has been done with a standard laboratory holographic setup, but it could be adapted and miniaturized to use this method in quality control.

## Figures and Tables

**Figure 1 polymers-14-01061-f001:**
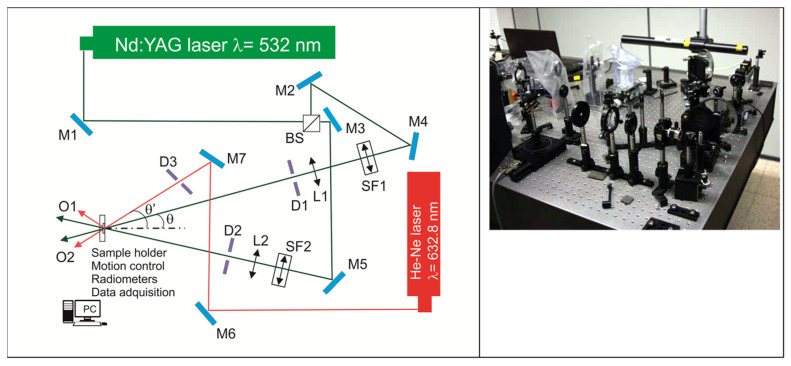
Holographic setup. BS: Beamsplitter, Mi: mirror, SFi: spatial filter, Li: lens, Di: diaphragm, Oi: radiometer, PC: data recorder.

**Figure 2 polymers-14-01061-f002:**
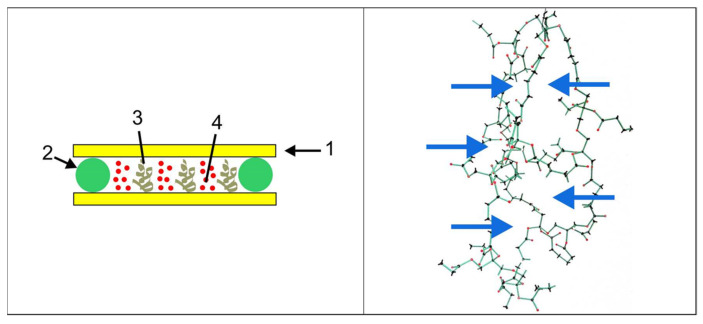
(**Left**) sample structure, 1: glass substrate, 2: microspacers, 3: exposed zones with polymer network, 4: unexposed zones with liquid crystal droplets. (**Right**) 3D polymer network from DPHPA. Arrows show holes in the network with a similar or higher size than an LC molecule.

**Figure 3 polymers-14-01061-f003:**
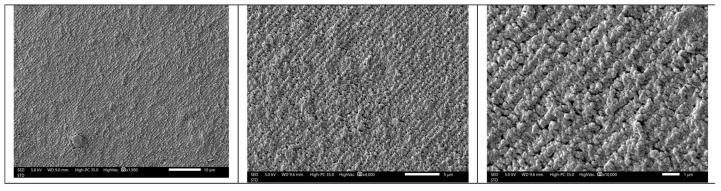
SEM images for a sample of HPDLC with peppermint oil (purity 80%).

**Figure 4 polymers-14-01061-f004:**
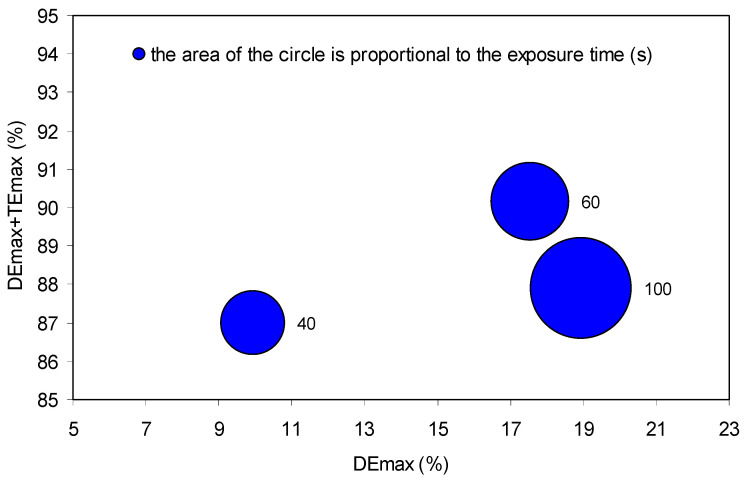
DEmax + TEmax versus DEmax for different exposure times during the hologram recording.

**Figure 5 polymers-14-01061-f005:**
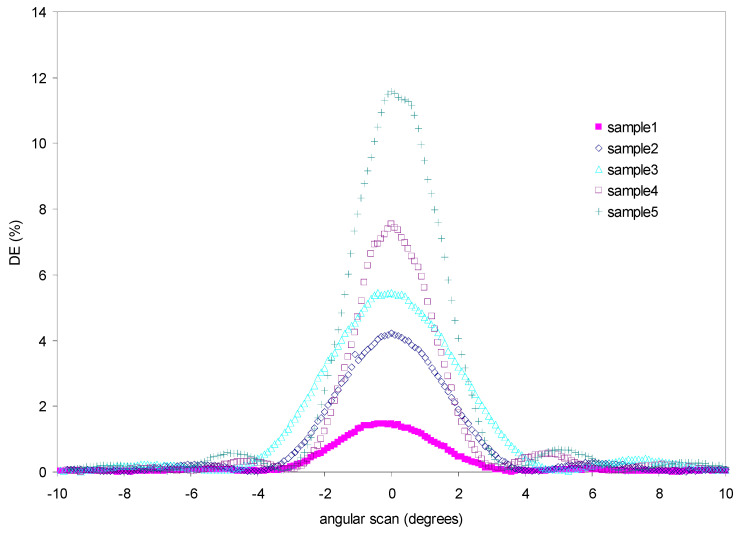
Diffraction efficiency versus angular scan for samples with different purity of peppermint oil.

**Figure 6 polymers-14-01061-f006:**
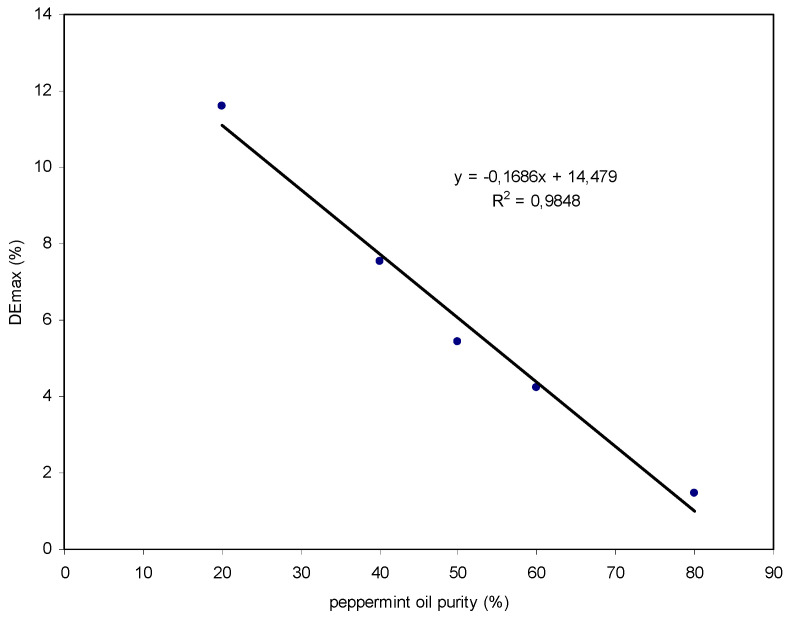
Maximum diffraction efficiency obtained as a function of the purity of the peppermint oil.

**Figure 7 polymers-14-01061-f007:**
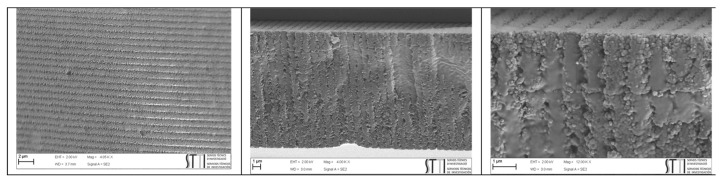
High-resolution SEM images for a sample of HPDLC with peppermint oil (purity 50%). Center and right images are taken by cross-sectional viewpoint.

**Figure 8 polymers-14-01061-f008:**
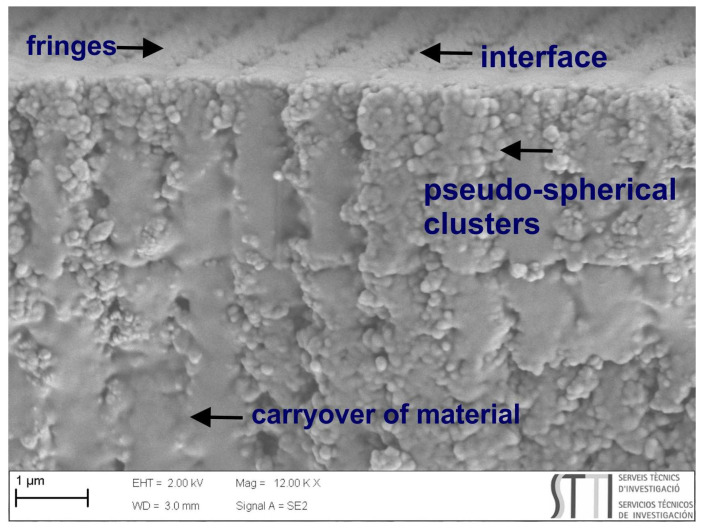
Enlargement of Figure 7 right showing aspects of the structure: interface between fringes, pseudo-spherical clusters that form the fringes, and areas with the carryover of material.

**Table 1 polymers-14-01061-t001:** Main components of the peppermint oil.

(+)-α-Pinene	Sabinene	(−)-β-Pinene	β-Myrcene
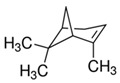	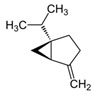	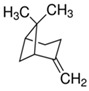	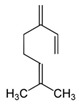
3-Octanol	R-(+)-Limonene	Eucaliptol	Isopregol
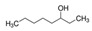	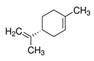	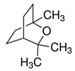	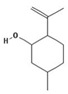
(−)-Isopulegol	Piperitone	(−)-Menthol	(−)-Menthone
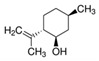	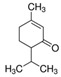	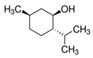	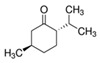
(+)-Isomenthone	Cis-3-Hexenyl isovalerate	(+)-Pulegon	(+)-2-Carene
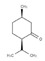	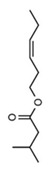	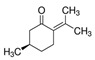	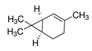

**Table 2 polymers-14-01061-t002:** Analysis of the composition of peppermint oil made by gas chromatography.

Peak Number	TR (min)	Area Component	Area (%)	Qual (%)
1	7.44	α-Pinene	0.73	97
2	8.54	Sabinene	0.34	91
3	8.62	β-Pinene	0.86	94
4	9.02	β-Myrcene	0.38	91
5	9.15	3-Octanol	0.39	83
6	10.11	R-(+)-Limonene	4.54	94
7	10.18	Eucaliptol	4.43	98
9	13.53	Isopregol	1.67	99
10	13.79	Isomenthone	17.81	97
12	14.09	Menthone	11.57	96
13	14.38	Menthol (±)	45.8	91
14	14.45	Isopulegol	1.04	60
15	14.64	Menthol	0.68	91
16	16.08	Isoleric Acid	0.42	83
17	16.21	Pulegon	1.29	97
18	16.63	Piperitone	0.87	95
19	17.71	(+)-2-Carene	6.96	91

**Table 3 polymers-14-01061-t003:** Physical parameters of peppermint oil and triethyl citrate.

Physical Parameters	Peppermint Oil	Triethyl Citrate
Density 25 °C (g/cm^3^)	0.893–0.905	1.14
n_D_^20^	1.459–1.465	1.442
Bp (°C)	215	235 (150mmHg)
Color	Colorless to pale yellow	Colorless
Optical rotation	−18 to −32	0

**Table 4 polymers-14-01061-t004:** Composition of the HPDLC.

Component	Concentration (wt%)
DPHPA	52.77
QYPDLC-036	36.27
YET	0.05
NPG	0.47
OA	10.44

**Table 5 polymers-14-01061-t005:** Samples of HPDLC/adulterated peppermint oil composite with different purity of the peppermint oil.

Sample	Purity of Peppermint Oil (%)	DE Max (%)	V Peppermint Oil (μL)	V Triethyl Citrate (μL)
Sample 1	80	1.5	40	10
Sample 2	60	4.2	30	20
Sample 3	50	5.4	25	25
Sample 4	40	7.5	20	30
Sample 5	20	11.6	10	40

**Table 6 polymers-14-01061-t006:** DEmax obtained for samples of HPDLC/adulterated peppermint oil composite.

Sample	Purity of Peppermint Oil (%)	DE Max (%)
Sample 1	80	1.5
Sample 2	60	4.2
Sample 3	50	5.4
Sample 4	40	7.5
Sample 5	20	11.6

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
