# Peer review of "Adulterant Detection in Peppermint Oil by Means of Holographic Photopolymers Based on Composite Materials with Liquid Crystal"

_polymers, 2022, doi:10.3390/polym14051061_

Round 1

Reviewer 1 Report

In this paper authors used as holographic material a HPDLC photopolymer containing nematic liquid crystal and peppermint oil. The authors studied the influence of the oil about the polymerization and the holographic response. The composite containing oil adulterated with triethyl citrate affected a diffraction efficiency related to the purity of the oil,  so the authors suggested the possibility of developing a holographic analysis method for control of row materials.

The paper  is clearly presented and the conclusions are confirmed by experimental data. The conclusions are clear,  so I consider the manuscript may be considered for publication in this form.

Author Response

Thank you for reviewing the work and the comments made.

Reviewer 2 Report

  1. In my opinion, this article may attract the attention of readers in the optics or analytical chemistry field. It will be better to submit to other suitable journals. There is no obvious correlation between the performance and the polymer property.

  1. Introduction

Authors mentioned that “The areas exposed to light are polymerized and their refractive index is higher than that of adjacent areas not exposed to light and therefore not polymerized[16]”.

This is true for pure photopolymer systems where there are polymerized domains and unreacted monomer domains.

In PIPS, the liquid crystal (LC) molecules diffuse to unexposed areas in the diffraction grating where they remain as droplets in this study (page 7).

(a) Will the peppermint oil and triethyl citrate diffuse to unexposed areas in the diffraction grating where they remain as droplets?

(b) The nematic liquid crystal QYPDLC-036 has an ordinary refractive index n0= 1.520 which is higher than Photocurable compositions.

Is the statement “The areas exposed to light are polymerized and their refractive index is higher than that of adjacent areas not exposed to light and therefore not polymerized” still valid for this study.

  1. The refractive index of peppermint oil and triethyl citrate are 1.459-1.465 and 1.442, respectively.

Please provide the refractive index of the five solutions.

I believe that the refractive index of the five solutions will not have big differences. Please provide a reasonable explanation about why there are obvious changes in diffraction efficiency for the five solutions.

Author Response

Response to Reviewer 2

Comments and Suggestions for Authors:

 In my opinion, this article may attract the attention of readers in the optics or analytical chemistry field. It will be better to submit to other suitable journals. There is no obvious correlation between the performance and the polymer property.

We have added a paragraph in Introduction to avoid confusion between the operation of a pure photopolymer and a HPDLC. We have added text in Conclusion to clarify the interpretation of the results.

It is possible evaluate this method as an analytical chemistry procedure making experiments to add statistical data, sensitivity, etc. in a future work

    Introduction

Authors mentioned that “The areas exposed to light are polymerized and their refractive index is higher than that of adjacent areas not exposed to light and therefore not polymerized[16]”.

This is true for pure photopolymer systems where there are polymerized domains and unreacted monomer domains.

I agree with the reviewer. We have added a new paragraph in Introduction to highlight the difference between pure photopolymer and HPDLC.

In PIPS, the liquid crystal (LC) molecules diffuse to unexposed areas in the diffraction grating where they remain as droplets in this study (page 7).

In HPDLC, LC molecules diffuse to unexposed areas during the holographic recording. Therefore there is a relatively high concentration of LC in unexposed areas. In HPDLC with mint oil the results are not according with this.

(a) Will the peppermint oil and triethyl citrate diffuse to unexposed areas in the diffraction grating where they remain as droplets?

After polymerization, LC remain as droplets inside the unexposed areas as a microemulsion in the monomer solution. The hologram is exposed to white light to polymerize the monomer of these areas and to fix the droplets to the polymer network.

From the experiments performed no information can be obtained as to whether the triethyl citrate/non-reactive components from mint oil diffuse together with the LC, but it is indifferent because their refractive indices are similar to the ordinary index of the LC and that of the monomer and the polymer.

In HPDLC, the LC diffusion affects refractive index modulation much more than monomer polymerization. But in HPDLC with mint, DEmax decreases when there is more mint oil and this is directly related to the fact that LC can not diffuse well to the unexposed areas.

(b) The nematic liquid crystal QYPDLC-036 has an ordinary refractive index n0= 1.520 which is higher than Photocurable compositions.

Yes, and the average refractive index is also higher

Is the statement “The areas exposed to light are polymerized and their refractive index is higher than that of adjacent areas not exposed to light and therefore not polymerized” still valid for this study.

 This statement is true for a pure photopolymer but not for HPDLC. We have added a new paragraph in Introduction to highlight the difference between pure photopolymer and HPDLC.

    The refractive index of peppermint oil and triethyl citrate are 1.459-1.465 and 1.442, respectively.

Please provide the refractive index of the five solutions.

I believe that the refractive index of the five solutions will not have big differences.

Yes, the 5 solutions have a similar refractive index

Please provide a reasonable explanation about why there are obvious changes in diffraction efficiency for the five solutions.

The changes in the diffraction efficiency for the different solutions are due to the different diffusion of LC to the unexposed areas influenced by differences in the size of the holes of the polymer network obtained in the polymerization.

Larger size of the holes in the polymer network formed implies less diffusion of LC to unexposed areas (same concentration of LC in exposed areas and unexposed areas) and therefore low index modulation and low diffraction efficiency

Changes made in the text:

-A new paragraph has been added to the end of the introduction

-We have added text in Conclusion to clarify the interpretation of the results.

-Axis x has been corrected in Figure 5. It is angular scan (degrees)

 -Acknowledments have been updated

Reviewer 3 Report

This paper presents a possibility of developing a holographic chemical analysis method (HPDLC) for quality control of raw materials such as the peppermint essential oil. The results and discussions concerning the purity of the essential oils in this paper are important for the presence of adulterate oils. However, this paper is not well written especially for the advantages of using this method. I cannot recommend publishing an article in its current form in Polymers until the author clarifies at least the points listed below.

1) Sometimes period "." is disappeared after blanket "]" at the end of sentence.

2) In page 4, at the caption of table 2, there are no description about "i".

3) In page 5, the author described that they prepared HPDLC/adulterated peppermint oil composite is prepared mixing 2.080 g of HPDLC with 0.469 ml of adulterated peppermint oil nevertheless they prepared 0.050 ml of adulterated peppermint oil in the table 5. What is the actual amount of an adulterated peppermint oil in HPDLC composite for sample 1 to 5?

4) In page 6, it is worth to add the sample with pure peppermint oil and triethyl citrate. And add the value of physical properties such as densities, refractive indices, boiling points, color, and optical rotation to show the advantage of HPDLC method for purity checking is much greater than the comparing physical properties.

5) In page 7, in the section 7.1, the author should be clarifying the purity of peppermint oil of HPDLC sample of Figure 3.

6) In page 7, figure 3, the scale bar and number are too small for reading. It is better to magnify the scale caption.

7) In page 11, figure 7, the scale bar and number are too small for reading. It is better to magnify the scale caption. It is worth to explain if the center and right images are taken by cross-sectional viewpoint. It is worth to add the explanation about which area indicates holes of polymer network in SEM images.

8) In page 11, figure 7, the author should be clarifying the purity of peppermint oil of HPDLC sample of Figure 7 and add the explanation

Author Response

This paper presents a possibility of developing a holographic chemical analysis method (HPDLC) for quality control of raw materials such as the peppermint essential oil. The results and discussions concerning the purity of the essential oils in this paper are important for the presence of adulterate oils. However, this paper is not well written especially for the advantages of using this method. I cannot recommend publishing an article in its current form in Polymers until the author clarifies at least the points listed below.

1) Sometimes period "." is disappeared after blanket "]" at the end of sentence.

Corrected

2) In page 4, at the caption of table 2, there are no description about "i".

Corrected

3) In page 5, the author described that they prepared HPDLC/adulterated peppermint oil composite is prepared mixing 2.080 g of HPDLC with 0.469 ml of adulterated peppermint oil nevertheless they prepared 0.050 ml of adulterated peppermint oil in the table 5. What is the actual amount of an adulterated peppermint oil in HPDLC composite for sample 1 to 5?

The amounts are 200 ml of HPDLC and 50 ml of adulterated peppermint oil. This has been corrected in the text (section 5 after Table 4).

4) In page 6, it is worth to add the sample with pure peppermint oil and triethyl citrate. And add the value of physical properties such as densities, refractive indices, boiling points, color, and optical rotation to show the advantage of HPDLC method for purity checking is much greater than the comparing physical properties.

All samples in Table 5 contain peppermint oil and triethyl citrate. A sample only with pure peppermint oil obtains DEmax=0. From a strict point of view, a sample containing pure essential oil without triethyl citrate cannot be compared with the rest. That's why it's not included.

To develop an analytical procedure with this holographic method is necessary to evaluate the physical constants that the reviewer says, specifically with each sample with different concentration of essential oil and triethyl citrate. Moreover it would be necessary to do statistical calculations to set the sensitivity and accuracy. All this has enough entity to develop it in a future work.

5) In page 7, in the section 7.1, the author should be clarifying the purity of peppermint oil of HPDLC sample of Figure 3.

It is a sample of HPDLC with peppermint oil (purity 80%). We have added this information to the text and figure caption.

6) In page 7, figure 3, the scale bar and number are too small for reading. It is better to magnify the scale caption.

We will upload the images to MDPI as supplementary material because a size equivalent to ½ text page is necessary to see the details and data well. If the scale caption is magnified, a large size is also necessary and the size of the white bar would change.

7) In page 11, figure 7, the scale bar and number are too small for reading. It is better to magnify the scale caption. It is worth to explain if the center and right images are taken by cross-sectional viewpoint. It is worth to add the explanation about which area indicates holes of polymer network in SEM images.

We will upload the images of Figure 7 to MDPI as supplementary material because it is difficult to achieve a satisfactory enlargement while maintaining the reference length of the figure caption and without it taking up a lot of size.

We have added an enlarged image of figure 7 in figure 8 to get as much information as possible. We have rewritten the information obtained from the figure.

8) In page 11, figure 7, the author should be clarifying the purity of peppermint oil of HPDLC sample of Figure 7 and add the explanation

Round 2

Reviewer 3 Report

This paper presents a possibility of developing a holographic chemical analysis method (HPDLC) for quality control of raw materials such as the peppermint essential oil. The revised manuscript is properly revised to be suitable for publication in its current form.